# Protocol for a systematic review assessing ambulatory vital sign monitoring impact on deterioration detection and related clinical outcomes in hospitalised patients

Carlos Areia ![ORCID],[1,2] Sarah Vollam ![ORCID],[1,2] Louise Young ![ORCID],[1,2] Christopher Biggs ![ORCID],[1,2] Marco Pimentel ![ORCID],[3] Mauro Santos ![ORCID],[2,3] Neal Thurley ![ORCID],[4] Stephen Gerry ![ORCID],[5] Lionel Tarassenko ![ORCID],[2,3] Peter Watkinson ![ORCID][1,2,6]

For numbered affiliations see end of article.

**Correspondence to**
Carlos Areia;
carlos.morgadoareia@ndcn. ox.ac.uk

## ABSTRACT

**Introduction** Ambulatory monitoring systems (AMS) can facilitate early detection of clinical deterioration, and have the potential to improve hospitalised patient outcomes. The objective of this systematic review is to assess the impact of vital signs monitoring on detection of deterioration and related outcomes in hospitalised patients using AMS, in comparison with standard care.

**Methods and analysis** A systematic search was conducted on 27 August 2020 in MEDLINE, Embase, CINAHL, Cochrane Database of Systematic Reviews, CENTRAL and Health Technology Assessment databases, as well as grey literature. Search results will be reviewed in accordance with the Preferred Reporting Items for Systematic Review and Meta-Analysis checklist for systematic reviews. Studies comparing the use of ambulatory monitoring devices against standard care for deterioration detection and related clinical outcomes in hospitalised patients will be included and further clinical and other outcomes will also be explored. Deterioration-related outcomes may include (but not limited to) unplanned intensive care admissions, rapid response team activation and unscheduled emergency interventions, as defined by the included studies. Two reviewers will independently extract study data and assess the quality and risk of bias of included studies. Where possible, a meta-analysis will be conducted and quantitative results presented. Alternatively, a narrative synthesis will be reported.

**Ethics and dissemination** Ethical approval is not required for this study as no primary data will be collected. This study is part of our virtual High Dependency Unit project and will be disseminated through peer-reviewed publications, public and scientific conference presentations.

**PROSPERO registration number** CRD42020188633.

## BACKGROUND

The utilisation of physiological early warning scoring (EWS) systems which monitor

### Strength and limitations of the study

► A potential limitation of this review is the focus on ambulatory monitoring systems that may limit the final number of included studies.
► This is the first systematic review assessing the effect of ambulatory monitoring devices on deterioration detection and related clinical outcomes in comparison with standard care.
► This review may provide an indication of the impact of ambulatory monitoring devices use in the hospital environment.
► We will review clinical trial registries and report the proportion of published studies against registered studies.

"standard" vital signs, including pulse rate, respiratory rate, blood pressure, oxygen saturations and temperature remains current practice, coupled with a graded response such as referral for a senior review or increasing monitoring frequency.[1] This frequency of observations is generally guided by the clinical condition of the patient. However, intermittent measurement of vital signs (including application of devices and documentation) can be time-consuming for healthcare professionals[2] and therefore, the desired monitoring frequency of vital signs is often not achieved.[3] This is identified as a limitation of intermittent monitoring systems, as they are dependent on the frequency of physiological monitoring observations[4]; of more concern is that even when ideal frequency is achieved, patients might deteriorate between observation sets.[5]

Higher risk patients are often continuously monitored, improving early detection

BMJ

of deterioration.[2] However, in the UK, continuous monitoring is not commonly used in the ward environment.[6] In a systematic review, Downey *et al* suggested that continuous vital sign monitoring outside the critical care setting may be feasible and has the potential to improve patient outcomes when compared with intermittent monitoring.[7 8] Another study suggested that implementing continuous monitoring in surgical wards was also cost effective.[8] However, clinical staff interviews indicate that despite the benefits and potential to increase the timely detection of patient deterioration, limitations in continuous vital sign monitoring technology can pose a barrier to implementation.[2] One of the main suggested reasons identified was invasiveness and restricted mobility.[6 7]

As a response to current limitations in healthcare monitoring, companies are extending the capabilities of commercially available wearable ambulatory vital sign monitoring systems.[9] These non-wired monitors may provide an alternative continuous monitoring system, affording patients more mobility, less discomfort, reduce nursing time and improve the early detection of abnormal physiological parameters.[10] Nevertheless, a major barrier to the clinical implementation of these systems is their unproven reliability, efficiency and data fidelity.[9] Another challenge is the potential detrimental effect of motion on the data derived from such monitoring systems, for example, motion is known to affect the accuracy of pulse oximetry readings.[11] This was seen throughout a clinically relevant range of measurements, with less accuracy at lower arterial oxygen saturations, which is clearly highly undesirable in clinical practice.

A recent systematic review and meta-analysis assessed the impact of multiparameter continuous non-invasive monitoring in hospital wards, suggesting that patients monitored in this way had a 39% decreased mortality risk when compared with those receiving intermittent monitoring; as well as a trend of reduced intensive care unit (ICU) transfer, rapid response team activation and length of stay at the hospital.[12] However, this review focused on non-invasive continuous monitoring and excluded neonatal, paediatric, obstetric, ICU and high-risk acute patients. Another recent review analysed the validation, feasibility, clinical outcomes and costs of 13 different wearable devices and concluded that these were predominantly in the validation and feasibility testing phases,[13] highlighting the lack of studies exploring clinical outcomes. However, this review focused on recent wearable devices, and only including studies between 2009 and 2019. Furthermore, while there are many devices claiming the ability to safely monitor patients at risk of deterioration,[14] evidence assessing the impact of ambulatory monitoring systems (AMS) in the ward environment remains inconclusive, limiting implementation and clinical use.[14]

## Aims and objectives

The objective of this systematic review is to assess the impact of vital sign monitoring in the deterioration detection and related clinical outcomes of hospitalised patients using AMS in comparison with standard care.

## METHODS AND ANALYSIS

This systematic review was registered with the International Prospective Register of Systematic Reviews on the 10 July 2020. This review will be conducted in accordance with the Preferred Reporting Items for Systematic Reviews and Meta-Analyses (PRISMA) checklist.[15] This protocol is in accordance with the PRISMA-Protocol statement checklist.[16]

### Methodological considerations
#### Inclusion and exclusion criteria
##### Types of studies

Randomised controlled trials (RCTs), interventional studies, observational studies (including case–control or cohort studies) and pilot studies will be included. Retrospective studies that comply with the proposed outcomes and eligibility criteria will be considered. To minimise publication bias, unpublished studies will also be included. There will be no date or language restrictions as every effort will be attempted to translate a non-English article.

Systematic, narrative and scoping reviews will be excluded, but we may use relevant reviews to facilitate identification of original data where appropriate. Diagnostic accuracy and validation studies purely assessing device accuracy or reliability will also be excluded. However, where possible for the included studies, we will extract appropriate diagnostic accuracy metrics as a secondary outcome.

##### Types of participants

We will include any studies conducted in the hospital environment. Although we are only including data from admitted patients, there are no restrictions in minimum length of stay or monitoring period. Studies conducted in healthy volunteers or non-hospitalised patients will be excluded.

##### Interventions

Studies that include non-invasive or minimally invasive ambulatory systems not as part of standard care will be included. As per its definition, AMS should not be restrictive or pose a barrier to patient mobility. To be included, AMS should monitor one or more vital sign (heart rate, respiratory rate, temperature, blood pressure or oxygen saturation) either continuously or intermittently (eg, sending streams of data every 5 min) and measurements should be automated, therefore requiring frequent manual input from the clinical staff. These systems can additionally:

▶ Include one or more ambulatory monitoring devices.
▶ Provide any other measurements that might be considered for analysis (such as ECG, photoplethysmography

(PPG) and accelerometry) as long as the device measures at least one vital sign.

### Comparators

For our primary outcome, we will include studies that compare the AMS to standard care. Standard ward care can include (but is not limited to) the following measurements:

► Heart rate: manual pulse rate count, spot checks using blood pressure machine/pulse oximeter, derived from PPG, ECG or arterial pressure waveforms.
► Respiratory rate: manual breathing rate count, derived from capnography or ECG waveform.
► Blood pressure: Blood pressure cuff reading, derived from arterial pressure waveform.
► Oxygen saturation: Pulse oximetry, arterial blood gas.
► Temperature: oropharyngeal, nasopharyngeal, rectal and skin thermometers.

If possible, we will describe the frequency of intermittent observations in studies using manual vital sign measurements as standard care.

If enough studies are available, we will do a subgroup analysis according to the type of standard monitoring (intermittent manual vital sign measurement or automated measurements) as these may vary between studies and wards (eg, ICUs using standard continuous wired monitoring).

As a secondary outcome, we will also include studies comparing different devices, if they comply with the eligibility criteria.

### Outcomes

We will include studies measuring deterioration detection and related clinical outcomes. If relevant, we will also include studies reporting other clinical outcomes in hospitalised patients similar to other reviews.[12] It is likely there may be a great variety of outcomes measurements not only on patient outcomes but also in devices' reliability and efficacy. Depending on the quantity and quality of papers fulfilling eligibility criteria and being included in the review, we may report a qualitative synthesis instead of a quantitative analysis.

#### Primary outcomes

We will aim to compare the AMS with standard care in regard to deterioration detection and related clinical outcomes metrics. We will be flexible, as we predict studies may use different outcomes and time frames. These can include (but are not limited to):

1. ICU admission.
2. Rapid response team activation.
3. Complications and adverse events.
4. Emergency/unscheduled interventions (eg, cardiopulmonary resuscitation, surgery and antibiotics).

#### Secondary outcomes

As a secondary outcome, we will report any other relevant patient-related outcomes (eg, mortality, length of stay and acute changes in therapy/medication). Similarly, these metrics are likely to vary between studies so there will be no predetermined criteria for inclusion at this point.

If enough data are available, we will also report device accuracy and reliability measurements, as well as the level of agreement between AMS and the standard care comparator (where available). We will also narratively explore patient experience and satisfaction with the AMS in the included studies.

#### Exploratory outcomes

We will narratively report alerting systems used and explore their impact on clinical outcomes for the included studies. This may include type of EWS, alert thresholds used for each vital sign, false alert rates and other relevant alert parameters or information.

We will also perform a clinical trial registry search and narratively report the proportion and characteristics of eligible registered studies against the number of results published (linking the respective registered studies end date and timelines with published reports).

### Information sources
#### Electronic searches

Relevant articles up to 27 August 2020 were identified through electronic searches MEDLINE Ovid (including epub ahead of print and in-process and other non-indexed citations), Embase (Ovid), CINAHL (EBSCO) and Cochrane Database of Systematic Reviews (Cochrane Library, Wiley), Cochrane Central Register of Controlled Trials (CENTRAL) (Cochrane Library, Wiley) and Health Technology Assessment database via the website (https://www.crd.york.ac.uk/CRDWeb/).

OpenGrey http://www.opengrey.eu/ was also searched for any unpublished grey literature. Given the popular culture/commercial applications of the technology, Google Scholar via https://scholar.google.com/ will also be searched. Additionally, we will also seek to identify eligible preprints using medRxiv via https://www.medrxiv.org/. If relevant papers are identified through included studies references, these may also be included.

Finally, we also searched trial registries such as ClinicalTrials.gov via https://clinicaltrials.gov/ and ISRCTN via https://www.isrctn.com/ for additional registered studies.

### Study records
#### Search strategy

The search strategy will be guided by a medical librarian (NT) who will provide support and guidance on search terms and keywords. Broad search terms will be used to capture the maximum number of publications in this area. MeSH terms and word truncations will be used where appropriate. An example search strategy is outlined in online supplemental appendix 1. Once the initial searches have been performed and a list of studies for inclusion has been agreed, we will conduct further searches if new relevant search terms are identified.

## Data extraction and management

Identified references will be downloaded to reference library software for the initial title and abstract screening; this will be handled using Rayyan software.[17] All searches will be saved for referencing, including a full list of papers and timeline of searches. The processing of each paper will be clearly documented throughout the screening and review process.

### Selection process

1. Once duplicates have been removed, two researchers will screen the titles and abstracts to exclude any evidently ineligible manuscripts using Rayyan software. Any disagreements will be included in the full-title review.
2. The full text of studies identified as potentially eligible following title and abstract screening will be assessed for inclusion by the two reviewers. Any disagreements in inclusion/exclusion of the article will be reviewed and discussed with a third reviewer.
3. Once articles are selected, hand-searching will be used to review their references to look for relevant articles not already identified. This is of particular importance as healthcare use of ambulatory monitoring devices or non-published articles may be identified using this method. If any relevant search terms are identified these will be included in a new search strategy following the above process.
4. The included articles will then be assessed by the two reviewers. Authors will be contacted by email for clarification or for further data wherever necessary. We will use a pilot data extraction form. The extraction output will be compared and a final data extraction form will be created. Extraction will include the author data, country, type of publication, date of publication, registration details, study design, setting, numbers of patients, eligibility criteria, missing data, AMS under study and type of measurement (intermittent/continuous), standard care comparator and type of comparator (intermittent/continuous), vital signs and other metrics measured, primary outcomes, secondary outcomes and other relevant metrics, alerting system (if there was one and how clinical staff was alerted), effect sizes and other estimates, population, demographics and cohort data, main conclusions, quality assessment and risk of bias assessment.

An overview of the selection process may be found in figure 1.

## Quality assessment

Methodological quality of RCTs will be assessed using the Jadad Scale.[18] For risk of bias and internal validity assessment, we will use the Cochrane risk of bias tool (RoB 2)[19] for randomised trials and the Newcastle Ottawa Scale for non-randomised studies.[20] Additionally, we will also use the mixed methods appraisal tool.[21]

Quality will be independently assessed and scores agreed by two reviewers; for significantly different scores and disagreements, a third reviewer will also conduct an independent quality assessment of the full-text. Scores will then be discussed and agreed between the three reviewers.

Where possible, visual assessment of funnel plots and Egger's regression will be used to assess publication bias. The grading of recommendations assessment, development and evaluation methodology will be used across studies to report the overall strength of the review as high, moderate, low or very low.[22]

## Data synthesis

### Quantitative or narrative approach to synthesising data

We will classify studies according to the interventions which are assessed. If we identify at least three studies assessing the effect of relatively homogeneous interventions, we will perform meta-analyses to combine the results. These meta-analyses will be on the basis of the outcome measures mentioned above. Therefore, it is likely that the effect measures of interest will be risk ratios or hazard ratios. If similar outcomes measures are reported using different effect measures, or time horizons, we will convert to a universal scale where possible, using standard methods. Effect measures will be combined using a random-effects meta-analysis. We will report the combined result (with 95% CI), a measure of between-study heterogeneity, and the 95% prediction interval. Where appropriate we will use sensitivity analyses to look at the effect of including/excluding non-randomised studies in the meta-analyses.

Meta analyses will be graphically represented using forest plots. An assessment of heterogeneity will be made (using both the $\chi^2$ test and the $I^2$ statistic) and estimates of effects will be extracted/calculated in accordance with the Cochrane Handbook for Systematic Reviews of Interventions V.6.0.[23] Statistical analyses will be carried out using R software.

If there are too few homogeneous studies to enable meta-analysis, results will instead be described through narrative synthesis. In this case, we will also explore different hospital environments and standard monitoring impact on the reported outcomes (eg, medical ward using standard intermittent monitoring as comparator or ICU using continuous monitoring) and other relevant subgroup analysis as outlined below.

### Subgroup analysis

If possible, we will explore heterogeneity between studies, through sensitivity and subgroup analyses. This will be undertaken only in studies with acceptable statistical methods and appropriate outcome reporting. These include the following:

1. According to different standard care comparator (eg, intermittent/manual or continuous monitoring).
2. Ward environment (ICU/surgical/medical).
3. Stratification according to each vital sign.
4. Stratification according to device used.
5. Subgroup analysis of non-adult population.

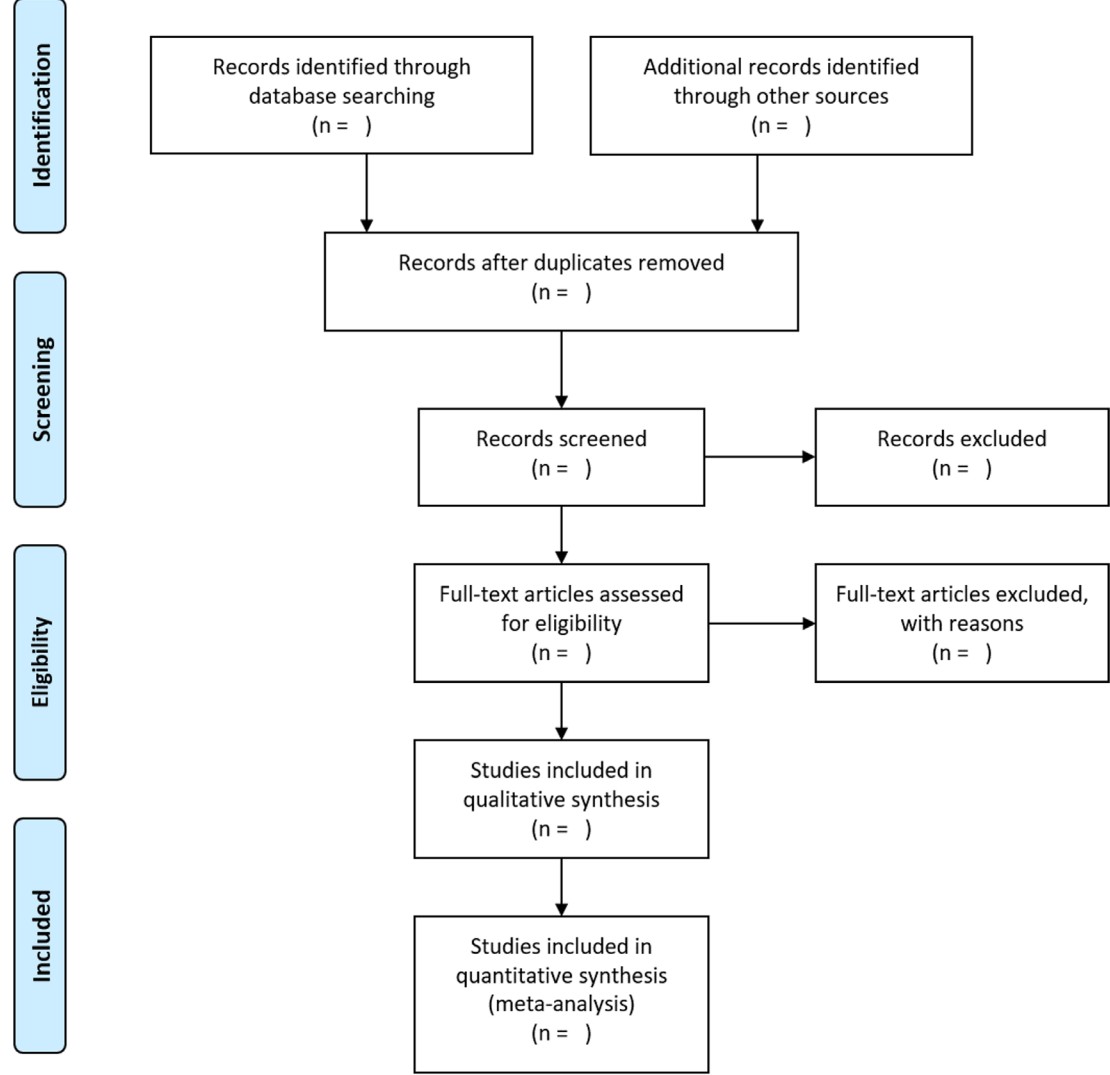

**Figure 1** Preferred Reporting Items for Systematic Reviews and Meta-Analyses flow diagram.

*Clinical trial registries analysis*

If any eligible studies are identified through searches of ClinicalTrialsGov and ISRCTN registries, we will perform a narrative synthesis exploring the proportion of registered studies against results published (considering the reported end date).

*Missing data*

Where missing or incorrect data are identified, corresponding authors will be contacted to clarify.

## Amendments to protocol

To ensure transparency, any change from this protocol will be documented and dated accordingly. No changes will be made to the main content of the protocol and any unexpected additional findings will be discussed in the systematic review.

## Patient and public involvement

This systematic review is part of the virtual High Dependency Unit project, aiming to implement ambulatory vital sign monitoring in the clinical environment. As this project develops, we will invite patients and members of the public to a group for ongoing support and feedback throughout the rest of the project. We have attended a number of local public engagement events, where members of the public showed interest and engaged in our vision of continuous wearable vital sign monitoring.

**Author affiliations**

[1]Critical Care Research Group, Nuffield Department of Clinical Neurosciences, University of Oxford, Oxford, UK
[2]National Institute for Health Research, Biomedical Research Centre, Oxford, UK
[3]Institute of Biomedical Engineering, Department of Engineering Science, University of Oxford, Oxford, UK
[4]Bodleian Health Care Libraries, University of Oxford, Oxford, UK
[5]Centre for Statistics in Medicine, Nuffield Department of Orthopaedics, Rheumatology and Musculoskeletal Sciences, University of Oxford, Oxford, UK
[6]Kadoorie Centre for Critical Care Research and Education, Oxford University Hospitals NHS Trust, Oxford, UK

**Contributors** CA: designed the study. CA, LY, SV and NT: designed the search strategy. NT: conducted databases search. CA, LY, SV, CB, MP, MS and SG: involved in data extraction and the assessment of methodological quality. PW and LT: final

approval of the study protocol. CA: wrote the initial draft of this manuscript and all authors contributed to and reviewed the final manuscript.

**Funding** The research was funded by the National Institute for Health Research (NIHR) Oxford Biomedical Research Centre (Award/Grant number: NIHR BRC-1215–20008). CA, SV, LY, CB, MS, LT and PW are supported by the NIHR Biomedical Research Centre, Oxford. The views expressed are those of the author(s) and not necessarily those of the NHS, the NIHR or the Department of Health.

**Competing interests** PW and LT report significant grants from the National Institute of Health Research (NIHR), UK and the NIHR Biomedical Research Centre, Oxford, during the conduct of the study. PW and LT report modest grants and personal fees from Sensyne Health, outside the submitted work. LT works part-time for Sensyne Health and has share options in the company. PW was Chief Medical Officer for Sensyne Health until March 2020 and holds shares in the company.

**Patient consent for publication** Not required.

**Provenance and peer review** Not commissioned; externally peer reviewed.

**ORCID iDs**
Carlos Areia http://orcid.org/0000-0002-4668-7069
Sarah Vollam http://orcid.org/0000-0003-2835-6271
Louise Young http://orcid.org/0000-0001-9094-1733
Christopher Biggs http://orcid.org/0000-0003-0348-5480
Marco Pimentel http://orcid.org/0000-0002-3696-8852
Mauro Santos http://orcid.org/0000-0002-1470-6966
Neal Thurley http://orcid.org/0000-0003-0770-7298
Stephen Gerry http://orcid.org/0000-0003-4654-7311
Lionel Tarassenko http://orcid.org/0000-0002-0118-1646
Peter Watkinson http://orcid.org/0000-0003-1023-3927

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
