## [Reviewer comments · BMJ Open]

ARTICLE DETAILS

TITLE (PROVISIONAL)	Protocol for a systematic review assessing ambulatory vital sign monitoring impact on deterioration detection and related clinical outcomes in hospitalised patients.
AUTHORS	Areia, Carlos; Vollam, Sarah; Young, Louise; Biggs, Christopher; Pimentel, Marco; Santos, Mauro; Thurley, Neal; Gerry, Stephen; Tarassenko, Lionel; Watkinson, Peter

VERSION 1 – REVIEW

REVIEWER	Goy, Jean-Jacques University & Hospital Fribourg, Cardiology
REVIEW RETURNED	07-Feb-2021

GENERAL COMMENTS	I would add as outcome an analysis of the false alarm of both standard system and new ambulatory monitoring system Would it be possible to also add the capability to detect significant arrhythmias with both systems With most of the new monitoring system it is reported how the patient accept it and how it prefer this new system to the old one. This could also be evaluated
---

REVIEWER	Nicholson, Margaret Liverpool Hospital, Intensive Care Unit
REVIEW RETURNED	08-Feb-2021

GENERAL COMMENTS	Standard care could be hourly or second hourly record of observations in some patient populations, if possible the frequency of standard care observations should be described. I look forward to your findings.
---

REVIEWER	Chan, P Eastern Health, Melbourne, ICU
REVIEW RETURNED	18-Feb-2021

GENERAL COMMENTS	I see no issues with this systematic review per
---

VERSION 1 – AUTHOR RESPONSE

Reviewer: 1

Dr. Jean-Jacques Goy, University & Hospital Fribourg

Comments to the Author:

I would add as outcome an analysis of the false alarm of both standard system and new ambulatory monitoring system

Response: Thank you for your insightful suggestion, we agree this would be really interesting to explore. We have now added this to the exploratory outcomes section. Hopefully, the included studies will have some information on the ambulatory monitoring systems false alert rates; we expect this to be more difficult in the standard care (as most studies will probably be using intermittent manual measurements).

Would it be possible to also add the capability to detect significant arrhythmias with both systems

Response: Thank you for your suggestion. Yes, I believe that will be covered inside the Primary outcome number 3 (Complications and adverse events). If there are sufficient studies we can explore this further.

With most of the new monitoring system it is reported how the patient accept it and how it prefer this new system to the old one. This could also be evaluated

Response: Thank you again for another insightful suggestion. Of course, this would be a great outcome to explore in the review. This has now been added to the secondary outcomes.

Reviewer: 2

Ms. Margaret Nicholson, Liverpool Hospital

Comments to the Author:

Standard care could be hourly or second hourly record of observations in some patient populations, if possible the frequency of standard care observations should be described. I look forward to your findings.

Response: Many thanks for your comment. Yes, we agree that should be described as we believe most studies will be using manual intermittent observations as standard care. This has now been added to the manuscript.

Reviewer: 3

Dr. P Chan, Eastern Health, Melbourne

Comments to the Author:

I see no issues with this systematic review per

Response: Thank you for taking the time to review our manuscript.